# A Simple Language Model for Task-Oriented Dialogue

**Ehsan Hosseini-Asl**
ehosseiniasl@salesforce.com
Salesforce Research

**Bryan McCann**
bmccann@salesforce.com
Salesforce Research

**Chien-Sheng Wu**
wu.jason@salesforce.com
Salesforce Research

**Semih Yavuz**
syavuz@salesforce.com
Salesforce Research

**Richard Socher**
rsocher@salesforce.com
Salesforce Research

## Abstract

Task-oriented dialogue is often decomposed into three tasks: understanding user input, deciding actions, and generating a response. While such decomposition might suggest a dedicated model for each sub-task, we find a simple, unified approach leads to state-of-the-art performance on the MultiWOZ dataset. SimpleTOD is a simple approach to task-oriented dialogue that uses a single, causal language model trained on all sub-tasks recast as a single sequence prediction problem. This allows SimpleTOD to fully leverage transfer learning from pre-trained, open domain, causal language models such as GPT-2. SimpleTOD improves over the prior state-of-the-art in joint goal accuracy for dialogue state tracking, and our analysis reveals robustness to noisy annotations in this setting. SimpleTOD also improves the main metrics used to evaluate action decisions and response generation in an end-to-end setting: inform rate by 8.1 points, success rate by 9.7 points, and combined score by 7.2 points.

## 1 Introduction

Conversational AI has been a long-standing area of exploration in computer science, and has gained more attention recently in both academia and industries with the current advances of neural approaches [15]. There are broadly two categories of dialogue. Open-domain dialogue systems focus on making chit-chat, open-ended conversations with humans more natural and engaging. They are usually trained end-to-end using large-scale data from social media [1, 42]. Task-oriented dialogue (TOD) systems accomplish a goal described by a user in natural language. They often use a pipeline approach [46, 58]. The pipeline requires natural language understanding (NLU) for belief state tracking, dialogue management (DM) for deciding which actions to take based on those beliefs, and natural language generation (NLG) for generating responses [50].

Traditionally, each component of task-oriented dialogue systems is trained independently with different supervision. The NLU module is trained on domain and intent labels. The DM module employs dialogue belief and dialogue act labels. The NLG module accesses templatized or natural responses. The modular dependencies of these components can lead to error propagation when information is not provided to subsequent modules in the pipeline [27]. For example, many systems do not consider the entire dialogue history at every turn, but rather rely on the NLU module to pass belief states reliably to following module components [60].

We propose recasting task-oriented dialogue as a simple, causal (unidirectional) language modeling task. We show that such an approach can solve all the sub-tasks in a unified way using multi-task

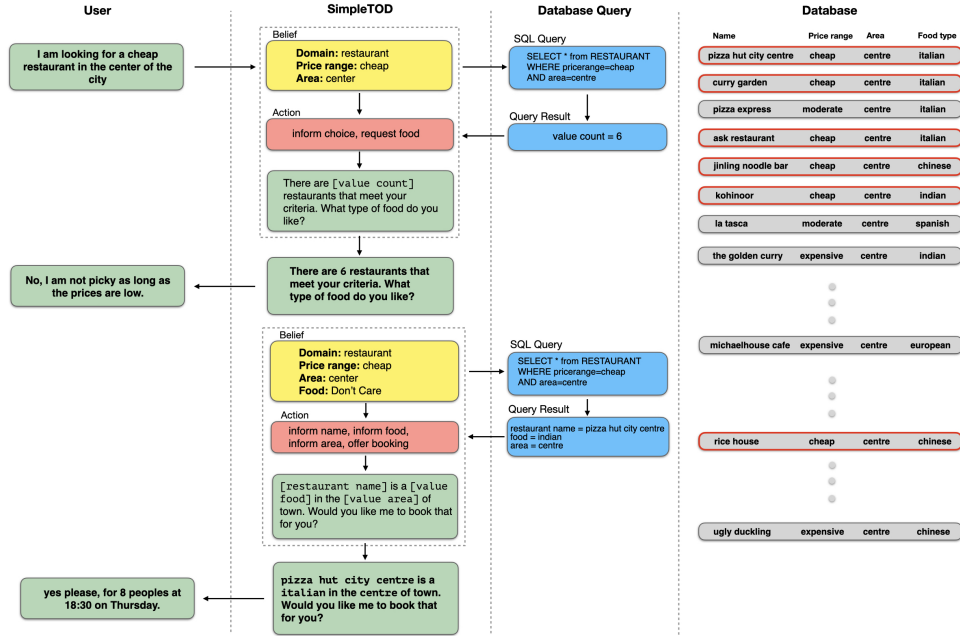

Figure 1: SimpleTOD is a simple approach to task-oriented dialogue that uses a single causal language model to generate all outputs given the dialogue context and retrieved database search results. The delexicalized response can then be lexicalized into a human-readable response by using information from the belief state and DB search results.

maximum likelihood training. The proposed Simple Task-Oriented Dialogue (SimpleTOD) approach enables modeling of the inherent dependencies between the sub-tasks of task-oriented dialogue, by optimizing for all tasks in an end-to-end manner. SimpleTOD also opens the path towards fully leveraging large language models such as GPT-2 [39] for task-oriented dialogue. The success of SimpleTOD demonstrates a strong connection between the implicit language understanding in the open domain required of high-quality causal language models and the kind of understanding required for a full task-oriented dialogue system.

Evaluation results demonstrate the advantages of SimpleTOD. It achieves 55.76 joint goal accuracy on MultiWOZ, which surpasses all prior work for the dialogue state tracking (i.e. belief state tracking) sub-task. In the setting closest to testing a full task-oriented dialogue system, in which belief states and action decisions are generated rather than retrieved from an oracle, SimpleTOD performance surpasses prior work on each individual action and response generation metric (+8.1 inform rate, +9.7 success rate).

The contributions of this work are summarized as follows:

- SimpleTOD – a state-of-the-art generative model for dialogue state tracking (DST).

- SimpleTOD is also the first model to achieve state-of-the-art performance for dialogue state tracking, action decisions, and response generation metrics together in an end-to-end setting.

- Analysis showing SimpleTOD is a robust dialogue state tracker in the presence of noisy-labeled annotations.

- Ablations showing the importance of *user/system* and *endof(segment)* tokens.

- Ablations showing the importance of pre-training that also show larger versions of Simple-TOD are not always better for end-to-end MultiWOZ.

- A list of discovered noisy annotations in MultiWOZ 2.1 alongside a cleaned version of the test set, code for training and evaluation, are provided at `https://github.com/salesforce/simpletod`

## 2  Related Work

**Task-Oriented Dialogue**   Most works on task-oriented dialogue focus on a specific module and evaluates only for that module. These components include understanding user intent via intent detection [26], tracking the constraints imposed by the user via dialogue state tracking [20, 32, 41, 34, 53, 59, 64, 8, 19], determining system actions via dialogue policy [51], and using dedicated response generation components [49].

Some recent works have started to bridge multiple sub-tasks by connecting modules together and evaluating in settings that hand off generated results from one module to another. Chen et al. [9] proposed a joint action-response generation using oracle dialogue states. Peng et al. [37] used GPT-2 to learn a response generator conditioned on oracle dialogue acts which did not evaluate on dialogue state tracking.

**Towards End-to-End Task-Oriented Dialogue**   Dependencies between these independent modules make pipeline approaches vulnerable to error propagation across components [28]. Recent approaches have increasingly shifted towards end-to-end solutions, which aim to reduce human effort and task-specific design. Several works used both dialogue history and knowledge bases as input and optimized neural encoder-decoder models to generate or retrieve system responses without modular supervision [13, 62, 29, 54, 56]. Some systems are mostly end-to-end, but still need to call out additional APIs or skip intermediate tasks like dialogue state tracking [5]. Others have incorporated additional supervision and trained in multi-task settings. Lei et al. [24] and Shu et al. [45] incorporated dialogue state tracking and jointly trained with response generation using a sequence-to-sequence approach. Liu et al. [28] proposed a hybrid imitation and reinforcement learning method, by jointly learning a policy for dialogue management with response generation. Wen et al. [50], Liang et al. [25] trained language understanding, dialogue state tracking, and dialogue policy modules with a shared encoder. Many other works fall somewhere in between by jointly training some tasks. Neelakantan et al. [33] modeled dialogue management and response generation jointly, incorporating latent knowledge reasoning through attention without using belief states. Zhao et al. [63] proposed to model system actions as latent variables, inducing a latent action space with variational inference methods. Zhang et al. [60] proposed a domain-aware multi-decoder model (DAMD) using augmented dialogue data, which achieved state-of-the-art combined score for dialogue management and response generation on the MultiWOZ dataset. Although all these approaches have come closer to unifying the stack, none are as simple as SimpleTOD: treating all of task-oriented dialogue as a single sequence prediction problem, using a single model, trained with a single, joint, multi-task loss.

**Unsupervised pre-training for natural language processing**   Pre-training approaches for natural language processing focus on transferable representations for contextualized word vectors [30, 38], generative models [39, 23], or a combination of both [12, 57]. Variants of pre-trained, bidirectional Transformers like BERT [11] are often evaluated on classification tasks such as those in the GLUE benchmark [48] or span-based question answering tasks [40]. Unidirectional (causal) pre-trained language models such as GPT-2 [39] or CTRL [23] resemble the decoder from the original Transformer architecture [47]. They aim to learn a distribution for next-word prediction, which makes them particularly useful for tasks that require text generation. In dialogue, Zhang et al. [61] built on GPT-2 by further pre-training it on Reddit data for open-domain response generation. Henderson et al. [21] also pre-trained on Reddit data with a dual Transformer encoder for response selection. Bao et al. [3] used both Twitter and Reddit data to pre-train a Transformer model with discrete latent variables. Wu et al. [55] proposed a response selection model by pre-training BERT model on multiple task-oriented corpora.

Budzianowski and Vulić [6] employed GPT-2 to leverage a pre-trained language model for dialogue response generation. They prepended dialogue context with belief state and DB search results. Their evaluation is only for context-to-response generation, and it is not applicable to the end-to-end setting. Wu et al. [56] used GPT2 in an alternating roles for the user and system, without using belief state or action annotation. It achieved better results than its predecessor on context-to-response generation (dialogue policy), but it requires oracle belief states to evaluate inform and success rate which is not applicable to end-to-end evaluation. It is also not designed for dialogue state tracking.

Ham et al. [17] fine-tuned GPT-2 on the MultiWOZ dataset and achieved lower performance on dialogue state tracking and end-to-end evaluation compared to the previous single-task and modularized models. They employed delimiter tokens for different segments: <usr>, <sys>, <ds> and <sa>. However, this differs from our tokenization, which is based on choosing semantic words which help the model to learn the semantic role of each segments – belief state, action, and response – and end of segment tokens such as <|endofbelief|>. They also made use of token-type embeddings for user and system sequences, whereas our model does not use this type of embedding layer. Their proposed model achieved lower performance than previous baselines for dialogue state tracking and in the end-to-end evaluation, whereas we outperform all previous models.

Peng et al. [36] proposed a GPT2-based model for end-to-end training (SOLOIST) by removing actions from the input sequence, and used token-type embedding for user and system responses as well. It is additionally pretrained on seven more dialogue datasets before fine-tuning on MultiWOZ. They used data augmentation during training using contrastive learning, similar to DAMD [60], where they combined dialogue context and belief states with a negative response and used the final token for classification to improve end-to-end performance. However, SOLOIST did not report end-to-end performance without pretraining on other dialogue datasets. Their "w/o pretraining" setting is only evaluated in a low resource setting, and they do not evaluate on dialogue state tracking as well.

In summary, our proposed model is much simpler than previous language model based approaches, in terms of (1) input sequence definition, (2) embedding layers, (3) training algorithm, and (4) pretraining, which make it easy to reproduce the results, and outperformed previous models on DST and end-to-end setting.

## 3 Methods

This section describes task-oriented dialogue, how we frame it for SimpleTOD, the model architecture, training details, dataset details, and evaluation metrics.

### 3.1 Task-Oriented Dialogue

Task-oriented dialogue (TOD) is evaluated on three sub-tasks: dialogue state (belief state) tracking, dialogue management (action/decision prediction) and response generation. This decomposition has made it possible to create dedicated models for each sub-task, which is the dominant approach. By contrast, we explore the possibility of using a single-model, end-to-end approach, SimpleTOD.

Dialogues consist of multiple turns. In a turn $t$, the user provides input $U_t$ and the system generates a response $S_t$. To generate a response during inference, SimpleTOD reads all previous turns as context, $C_t = [U_0, S_0, \ldots, U_t]$. It generates a belief state $B_t$,

$$B_t = \text{SimpleTOD}(C_t) \tag{1}$$

which is a list of triplets recording values for slots in a particular domain: *(domain, slot_name, value)*. This belief state is used to query a database for information. The database search returns rows from the database that satisfy the conditions of the belief state. The rows returned can later be used to lexicalize the response (filling in generated placeholders), but SimpleTOD only takes as input the aggregated database search results, $D_t$. $D_t$ includes how many rows were returned and, depending on the experimental setting, whether booking status information. SimpleTOD then conditions on $C_t$, $B_t$, and $D_t$ concatenated together as a single sequence to decide actions, $A_t$.

$$A_t = \text{SimpleTOD}([C_t, B_t, D_t]) \tag{2}$$

These actions are generated as another list of triplets: *(domain, action_type, slot_name)*. A delexicalized response $S_t$ is generated conditioned on all prior information concatenated as a single sequence.

$$S_t = \text{SimpleTOD}([C_t, B_t, D_t, A_t]) \tag{3}$$

When combined with information from the belief state and database search results, the response can be lexicalized to recover human readable response text. Figure 2 depicts training of SimleTOD and generation during inference.

## 3.2 Causal Language Modeling

A single training sequence consists of the concatenation $x^t = [C_t; B_t; D_t; A_t; S_t]$ [1], allowing us to model the joint probability over the sequence $x^t$. Given example sequences of the form $x = (x_1, \ldots, x_n)$ where each $x_i$ comes from a fixed set of symbols, the goal of language modeling is to learn $p(x)$. It is natural to factorize this distribution using the chain rule of probability [4] and train a neural network with parameters $\theta$ to minimize the negative log-likelihood over a dataset $D = \{x^1, \ldots, x^{|D|}\}$ where sequence $x^t$ has length $n_t$:

$$p(x) = \prod_{i=1}^{n} p(x_i | x_{<i}) \qquad \mathcal{L}(D) = -\sum_{t=1}^{|D|} \sum_{i=1}^{n_t} \log p_\theta(x_i^t | x_{<i}^t) \tag{4}$$

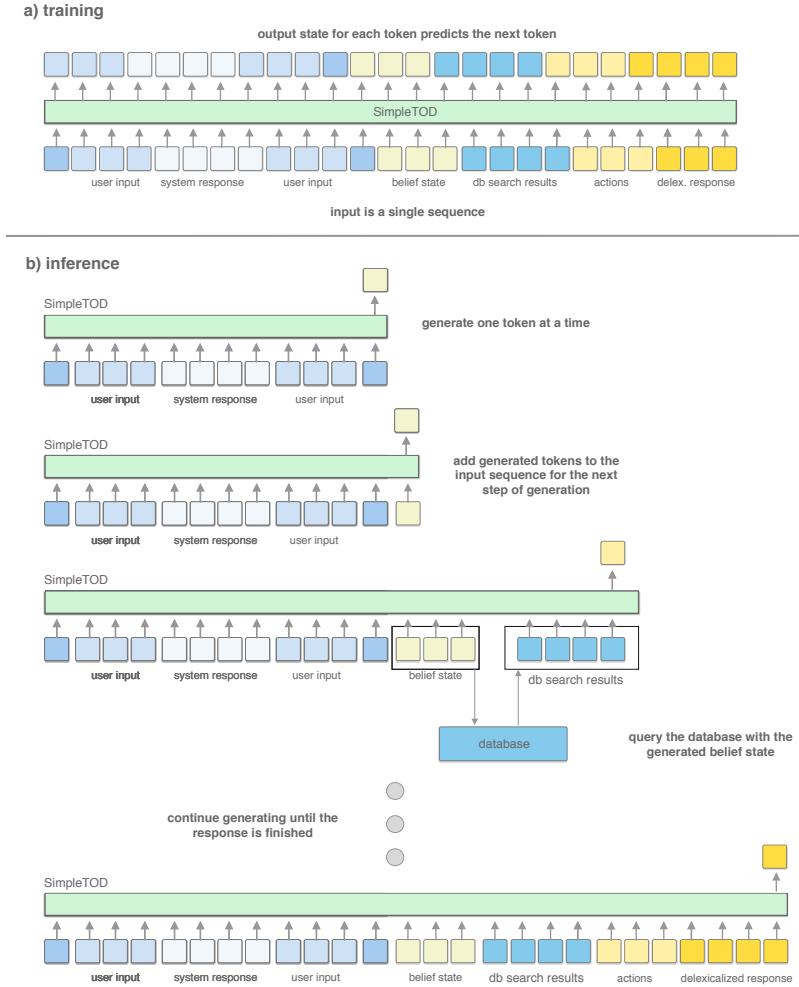

Figure 2: SimpleTOD is a simple approach to task-oriented dialogue that approaches all of task-oriented dialogue as a single sequence generation problem, querying a database for necessary information.

## 3.3 Architecture

We train a variant of the Transformer [47] to learn these conditional distributions. A sequence containing $n$ tokens is embedded as a sequence of $n$ vectors in $\mathbb{R}^d$. Each vector is the sum of a

learned token embedding and a sinusoidal positional embedding. The sequence of vectors is stacked into a matrix $X_0 \in \mathbb{R}^{n \times d}$ and processed by $l$ attention layers. The $i$th layer consists of two blocks, each preserving model dimension $d$. The first block uses multi-head attention with $k$ heads. A causal mask precludes attending to future tokens:

$$\text{Attention}(X, Y, Z) = \text{softmax}\left(\frac{\text{mask}(XY^\top)}{\sqrt{d}}\right)Z$$

$$\text{MultiHead}(X, k) = [h_1; \cdots; h_k]W_o$$

$$\text{where } h_j = \text{Attention}(XW_j^1, XW_j^2, XW_j^3)$$

The second block uses a feedforward network with ReLU activation that projects inputs to an inner dimension $f$. This operation is parameterized by $U \in \mathbb{R}^{d \times f}$ and $V \in \mathbb{R}^{f \times d}$:

$$FF(X) = \max(0, XU)V$$

Each block precedes core functionality with layer normalization [2, 10] and follows it with a residual connection [18]. Together, they yield $X_{i+1}$:

| Block 1 | Block 2 |
|---|---|
| $\bar{X}_i = \text{LayerNorm}(X_i)$ | $\bar{H}_i = \text{LayerNorm}(H_i)$ |
| $H_i = \text{MultiHead}(\bar{X}_i) + \bar{X}_i$ | $X_{i+1} = \text{FF}(\bar{H}_i) + \bar{H}_i$ |

Scores are then computed from the output of the last layer:

$$\text{Scores}(X_0) = \text{LayerNorm}(X_l)W_{vocab}$$

During training, these scores are the inputs of a cross-entropy loss function. During generation, the scores corresponding to the final token are normalized with a softmax, yielding a distribution for sampling a new token.

### 3.4   Training Details

The input to the model is tokenized with pretrained BPE codes [44] associated with DistilGPT2 [43], a distilled version of GPT-2 [39]. According to experimental results, Experiments for SimpleTOD use default hyperparameters for GPT-2 and DistilGPT2 in Huggingface Transformers[52]. Sequences longer than $1024$ tokens are truncated.

### 3.5   Dataset Details

We evaluate on the Multi-domain Wizard-of-Oz (MultiWOZ) [7], a large-scale, multi-domain dialogue dataset of human-human conversations. It contains 10438 multi-turn dialogues with 13.68 average turns, spanning over seven domains (restaurant, train, attraction, hotel, taxi, hospital, police). Police and hospital domains are excluded from evaluation, since they do not have valid/test splits. This leaves 30 domain-slot pairs for the remaining five domain with 4,500 possible values. SimpleTOD is trained on delexicalized system responses according to the pre-processing explained in [7]. Recently, [14] released MultiWOZ 2.1 which removes some noisy state values from dialogue state (belief state) tracking annotations. For dialogue state tracking evaluation, we used 2.1 version in order to compare to recent state-of-the-art methods. To the best of our knowledge, all prior work on action and response generation has evaluated on 2.0, so we include those results for direct comparison. But, we also include results for 2.1 so future work can compare to SimpleTOD on the improved version as well.

### 3.6   Evaluation Details

We follow the original MultiWOZ [7] guidance for all individual metrics and follow Mehri et al. [31] for the combined score. Joint goal accuracy is used to evaluate the performance of dialogue state tracking (i.e. belief state tracking). It measures the accuracy of the generated belief states as they compare to oracle belief states. Model outputs are only counted as correct when all the predicted values exactly match the oracle values. Action and response generation uses three metrics. The first two are inform and success rates. They are designed to capture how well the task was completed. Inform rate measures how often the entities provided by the system are correct. Success rate refers to how often the system is able to answer all the requested attributes by user. BLUE score [35] is used to measure the fluency of the generated responses. The combined score for action and response generation is computed as $(BLEU + 0.5 * (Inform + Success))$.

| Model | Decoder | Context Encoder | Extra Supervision | Joint Accuracy |
|---|---|---|---|---|
| TRADE[*] | Generative + Classifier | Bidirectional | - | 45.6 |
| DSTQA[**] | Classifier | Bidirectional | knowledge graph | 51.17 |
| DST-Picklist[*] | Classifier | Bidirectional | - | 53.3 |
| SST[*] | Generative | Bidirectional | schema graph | 55.23 |
| TripPy[†] | Classifier | Bidirectional | action decision | 55.3 |
| SimpleTOD[o] | Generative | Unidirectional | - | 55.72 |
| SimpleTOD[*] | Generative | Unidirectional | - | **55.76** |
| SimpleTOD[+] | Generative | Unidirectional | - | 57.47 |

Table 1: Evaluation of Dialogue State Tracking (DST) on MultiWOZ 2.1 using joint accuracy metric. [*] uses test label cleaning proposed by Wu et al. [53] and recommended by MultiWOZ authors. [†] uses label normalization and equivalent matching proposed in Heck et al. [19]. [**] uses the cleaning of [*] models plus additional accounting for label variants. [+] performs cleaning of Type 2 and partial cleaning of Type 4 noisy annotations as outlined in Section 5, which is currently non-standard and so left unbolded. [o] no label-cleaning.

# 4 Experimental Results and Discussion

**SimpleTOD is a Unified System for Task-Oriented Dialogue**    SimpleTOD is, to the best of our knowledge, the first system that generates state-of-the-art results judged according to dialogue state tracking as well as end-to-end metrics for action and response generation for MultiWOZ.

## 4.1 Dialogue State Tracking

Table 1 compares the joint goal accuracy to previous methods. We compare to TRADE [53], DSTQA [64], DST-Picklist [59], SST [8], and TripPy [19]. All previous models propose a bidirectional encoder to learn a better representation of the dialogue context, but SimpleTOD uses a unidirectional (causal) decoder and no additional bidirectional encoder. It also makes no use of extra supervision. It nonetheless achieves state-of-the-art.

Many models use some form of test-label cleaning. TRADE, DSTQA, DST-Picklist, and SST use the script proposed by Wu et al. [53][2]. DSTQA also accounts for label variations that would have originally been considered incorrect. TripPy apply their own format normalization, typo corrections, and process for accounting for label variations. SimpleTOD achieves the best performance without any cleaning or normalization, simply on the raw, original annotations. Applying the script from Wu et al. [53] improves the result to 55.76. Analysis of further noisy annotation is presented in section 5. Further cleaning those annotations more accurately reflects performance at 57.47. We will release the list of noisy annotations that need to be fixed along with their corrections, but we reiterate that SimpleTOD does not need this cleaning to surpass prior methods.

## 4.2 Action and Response Generation

Table 2 and Table 3 demonstrate the effectiveness of SimpleTOD for action and response generation in the most realistic, fully end-to-end[3] setting – when models must generate belief states, actions, and responses. SimpleTOD targets replacing modularized and pipelined methods that evaluate different components evaluated with oracle information. For reference, oracle settings compare across a variety of settings against HDSA [9], ARDM [56], LaRL [63], PARG [16] can be found in the Supplementary Materials, but these comparisons are not essential for end-to-end contributions. In fact, SimpleTOD is state-of-the-art in the end-to-end setting compared to the only prior work, DAMD [60], without achieving state-of-the-art in settings that partially utilize oracle information. This highlights that partial, oracle evaluation does not reliably transfer to the end-to-end evaluation of full systems – only end-to-end evaluation accurately describes the performance of a full system.

| Model | Belief State | DB Search | Action | Inform | Success | BLEU | Combined |
|---|---|---|---|---|---|---|---|
| DAMD+augmentation | generated | oracle | generated | 76.3 | 60.4 | 16.6 | 85 |
| SimpleTOD (ours) | generated | oracle | generated | 78.1 | 63.4 | 16.91 | 87.66 |
| SimpleTOD (ours) | generated | dynamic | generated | 81.4 | 69.7 | 16.11 | 91.66 |
| SimpleTOD (ours) | generated | - | generated | **84.4** | **70.1** | 15.01 | **92.26** |

Table 2: Action and response generation on MultiWOZ 2.0 reveals that SimpleTOD, a single, causal language model, is sufficient to surpass prior work.

| Belief State | DB Search | Action | Inform | Success | BLEU | Combined |
|---|---|---|---|---|---|---|
| generated | oracle | generated | 79.3 | 65.4 | 16.01 | 87.36 |
| generated | dynamic | generated | 83.4 | 67.1 | 14.99 | 90.24 |
| generated | - | generated | 85 | 70.5 | 15.23 | 92.98 |

Table 3: Action and response generation on MultiWOZ 2.1 for SimpleTOD.

Prior work uses oracle DB Search results as supervision during training and as input during inference. We include directly comparable experiments using oracle DB Search results. We also include experiments that completely ignore the DB Search results to show the surprising effectiveness of SimpleTOD without DB Search information. We also show a setting with dynamic DB Search results. In this setting, we train with the number of matched DB entries and compute this dynamically at inference from generated belief states. In all variations, SimpleTOD outperforms prior work.

DAMD [60] is the only prior work that has evaluated with generated belief states from dialogue state tracking during inference. We found in additional ablation experiments that we could increase scores for individual metrics like inform rate and success rate by training three separate SimpleTOD language models: one for dialogue state tracking, one for action generation, and one for response generation. However, the combined scores remained nearly identical to the full end-to-end, single model approach. For example, separating the models might improve inform rate, but hurt response generation measured by BLEU. Regardless, in this most realistic setting SimpleTOD achieves state-of-the-art on inform and success metric. SimpleTOD performs lower only on BLEU by 1.59 points, perhaps due to lack of action/response augmentation employed by DAMD.

**Regarding Oracle DB Search Results**   In the case where we dynamically compute partial DB Search results (number of entries matched only), the results are actually lower than ignoring them entirely. Using oracle DB information likewise leads to lower performance. The best result ignores DB Search results entirely. We have found that in some cases, the generated belief states conflict in some way with the information in the database. For example, there can be discrepancies between the two in the name of restaurants: 'pizza hut fenditton' in the target belief states but 'pizza hut fen ditton' in the database. We have consulted with the authors of the dataset, but there is currently no course of action planned to remedy this.

## 5   Analysis and Further Discussion

**The Role of Special Tokens**   Table 4 evaluates SimpleTOD with different special tokens used to identify components of the input corresponding to different sub-tasks. Analysis revealed that without end tokens, SimpleTOD tended to generate much longer belief state, action, and response generations. Even more important is clearly differentiating user and system text for SimpleTOD.

**Pre-training**   Table 5 highlights the importance of initializing SimpleTOD with pre-trained weights. A major advantage of recasting as single sequence prediction is the ability to leverage the understanding learned by these pre-trained models in the open-domain setting.

**Robustness to Noisy Annotations**   To understand the source of dialogue state tracking errors, we investigated MultiWOZ 2.1 annotations in depth. In the process, we have defined four primary types of noisy-labels that could be considered mis-annotations:

1. User provided multiple options, but context does not provide sufficient information to determine the true belief state.

| End token | User/System token | Joint Acc | Inform | Success | BLEU | Combined |
|-----------|-------------------|-----------|--------|---------|------|----------|
| No | No | 16.79 | 33.8 | 10.6 | 4.53 | 26.73 |
| Yes | No | 21.5 | 54.5 | 41.2 | 9.48 | 57.33 |
| No | Yes | 22.22 | 61.9 | 52.7 | 9.57 | 66.87 |
| Yes | Yes | 55.76 | 85 | 70.5 | 15.23 | 92.98 |

Table 4: Ablations on MultiWOZ 2.1 comparing the presence and absence of different special tokens when representing TOD as a single sequence. Performance on all metrics drops without *<endof(segment)>* and *<user/system>* tokens.

| Layers | Pretrained | Joint Acc | Inform | Success | BLEU | Combined |
|--------|-----------|-----------|--------|---------|------|----------|
| 6 | Random | 16.45 | 63.5 | 49.6 | 6.34 | 62.89 |
| 6 | DistilGPT2 | 54.54 | 85 | 70.5 | 15.23 | 92.98 |
| 12 | Random | 20.17 | 58.7 | 37.4 | 8.9 | 59.65 |
| 12 | GPT2 | 55.76 | 88 | 61.7 | 15.9 | 90.75 |

Table 5: Ablations on MultiWOZ 2.1 comparing the importance of pretraining. Recasting as single sequence prediction enables fully leveraging pre-trained models for the language understanding they have gathered in an open-domain setting.

2. Belief state is not labeled, but context provides sufficient information.

3. Belief state is labeled, but context lacks necessary information.

4. Belief state value is misspelled according to the context information.

Together experimental results and this analysis indicate that SimpleTOD can track dialogue state and generate the correct output even in the presence of noisy labels. Concrete examples of noisy-labeled annotation in MultiWOZ can be found in the Supplementary Materials. All mis-annotated examples along with all code for replication are provided [4].

**Decoding**    Initialized from pre-trained weights, SimpleTOD does not need to employ an advanced, more costly decoding strategy such as beam search, diverse beam search, and top-k sampling as opposed to HDSA [9] and DAMD [60]. Our results are reported with simple greedy decoding. In initial experiments, we also tried nucleus sampling [22], but we found it degraded performance. This relates to the observations in Keskar et al. [23] around controllable generation: when precision is required, sampling from the distribution is inherently less reliable than greedily sampling.

**Full Dialogues, Multiple Turns, and Long Contexts**    In further analysis, we found that Simple-TOD accurately tracks dialogue state over multiple turns and long contexts. In some cases, earlier belief state errors are remedied later on when additional turns provide increased context. Examples of full dialogues and those with many turns or especially long context can be found in Supplementary Materials, but we do not consider this further analysis as a primary contribution listed for the work.

## 6    Conclusion

We explored a simple approach to task-oriented dialogue (SimpleTOD) that uses a single, causal language model. To do this, during training we treat all inputs for dialogue state tracking, action and response generation as a single sequence to the model. SimpleTOD can then directly leverage pre-trained models like GPT-2 to transfer language understanding from open-domain settings where data is more readily available. Empirical results on the multi-domain dialogue dataset (MultiWOZ) showed that the proposed approach outperformed all prior methods in dialogue state tracking as well as in action and response generation in the end-to-end setting. We found that the pre-trained weights were essential, but to leverage these weights fully we had to guide the system with special tokens that mark user and system responses as well as different portions of the sequence related to different sub-tasks. We found that SimpleTOD was effective at tracking dialogue state over long context with many turns and required no more than greedy decoding to achieve new state-of-the-art results despite noisy annotations. We hope that these results and the code, models, and discovered noisy annotations will encourage further exploration of simple, unified approaches for dialogue systems.

# 7 Broader Impact

This work may have implications for the simplification of conversational agents. In the narrow sense, this work addresses task-oriented dialogue, but similar results might also hold for open-domain conversational systems. If so, the improvement of these systems and easier deployment would amplify both the positive and negative aspects of conversational AI. Positively, conversational agents might play a role in automating predictable communications, thereby increasing efficiency in areas of society that currently lose time navigating the multitude of APIs, webpages, and telephonic systems that are used to achieve goals. Negatively, putting conversational agents at the forefront might dehumanize communication that can be automated and might lead to frustration where human agents could provide more efficient solutions – for example, when predicted solutions do not apply. These consequences are not specific to this work, but should be considered by the field of conversational AI more broadly.

## Footnotes

[1]During inference, $D_t$ comes from a database. See Sec. 4 for experimental results revealing that it can be advantageous to exclude this from training.

[2] `https://github.com/jasonwu0731/trade-dst/blob/master/utils/fix_label.py`

[3] The term "end-to-end" is overloaded in the literature. Evaluation that does not use oracle belief states, actions, or response is considered end-to-end even when the system itself is not trained end-to-end. SimpleTOD is trained end-to-end and achieves state-of-the-art in end-to-end evaluation.

[4]`https://github.com/salesforce/simpletod`

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
