[Supplementary Material]

# A Input Representation and Method Overview

As described in Section 3, a single training sequence consists of the concatenation of context $C_t$, belief states $B_t$, database search results $D_t$, action decisions $A_t$, and system response $S_t$. A schematic overview of each segment is shown in Table 6 together with special tokens marking transition points. SimpleTOD is optimized by minimizing the negative likelihood over the joint sequence $x^t = [C_t; B_t; D_t; A_t; S_t]$. The output state associated with each input token is used to predict the next token, see Figure 1a.

During inference, SimpleTOD generates this sequence token by token, but we stop after belief states are generated to query from a database. The outputs of the database are summarized and concatenated to the end of the input sequence and generation resumes token by token. This results in a delexicalized response, see Figure 1b. This response can then be lexicalized by replacing slots and values with information from the database results. This process is described more formally in the equations of Section 6.

| | |
|---|---|
| Context | [context] [user] *user input* [system] *system response* ... [user] *user input* [endofcontext] |
| Belief State | [belief] *domain slot_name value, domain slot_name value, ...* [endofbelief] |
| DB Search | [db] *#_matches, booking_status* [endofdb] |
| Action | [action] *domain action_type slot_name, domain action_type slot_name, ...* [endofaction] |
| Response | [response] *system delexicalized response* [endofresponse] |

Table 6: A schematic representation of the different components of inputs/outputs in task-oriented dialogue. When training SimpleTOD, these are concatenated together into a single sequence.

# B SimpleTOD with Oracle information

This section reports the performance of SimpleTOD for action and response generation, in the presence of different oracle information, i.e. oracle belied and oracle action. These settings are not end-to-end as in the main text, and SimpleTOD is designed to be end-to-end. We report results in these settings for a complete understanding of SimpleTOD, but we note that only the end-to-end settings in the main text evaluate the full system together. In these oracle settings, other methods can outperform SimpleTOD, but this simply highlights the importance of end-to-end evaluation: there is a disconnect between performance with oracle information and performance without it. In practical use, oracle information is not available, and that is where SimpleTOD excels.

We report results in Table 7 for two different settings regularly employed in the literature. These settings are determined by how much oracle information is used. The first setting uses oracle belief states and oracle actions. The second uses oracle belief states, but requires the system to generate its own actions.

Note that all prior works use oracle DB Search results as supervision during training and as input during inference in all these settings. We include directly comparable experiments using oracle DB Search results for all settings. We also include experiments that completely ignore the DB Search results in all settings to show the surprising effectiveness of SimpleTOD without DB Search results.

The evaluation results on MultiWOZ 2.1, as shown in Table 8, also follow the same patterns as discussed in section 4.2. We provide these results for future comparisons on the improved version of the dataset.

| Model | Belief State | DB Search | Action | Inform | Success | BLEU | Combined |
|---|---|---|---|---|---|---|---|
| DAMD+augmentation | oracle | oracle | oracle | **95.4** | **87.2** | **27.3** | 118.5 |
| PARG | oracle | oracle | oracle | 91.1 | 78.9 | 18.8 | 103.8 |
| SimpleTOD (ours) | oracle | oracle | oracle | 93.4 | 83.2 | 17.78 | 106.08 |
| SimpleTOD (ours) | oracle | - | oracle | 92.3 | 85.8 | 18.61 | 107.66 |
| HDSA | oracle | oracle | generated | 82.9 | 68.9 | 23.6 | 99.5 |
| DAMD+augmentation | oracle | oracle | generated | **89.2** | 77.9 | 18.6 | 102.5 |
| ARDM | oracle | oracle | - | 87.4 | 72.8 | 20.6 | 100.7 |
| LaRL | oracle | oracle | generated | 82.78 | **79.2** | 12.8 | 93.79 |
| SimpleTOD (ours) | oracle | oracle | generated | 84 | 72.8 | 16.1 | 94.5 |
| SimpleTOD (ours) | oracle | - | generated | 88.9 | 67.1 | 16.9 | 94.9 |

Table 7: SimpleTOD results on MultiWOZ 2.0 using oracle information.

| Belief State | DB Search | Action | Inform | Success | BLEU | Combined |
|---|---|---|---|---|---|---|
| oracle | oracle | oracle | 92.8 | 84.5 | 18.9 | 107.55 |
| oracle | - | oracle | 92.6 | 86.1 | 17.67 | 107.2 |
| oracle | oracle | generated | 85.1 | 73.5 | 16.22 | 95.52 |
| oracle | - | generated | 89.6 | 68.6 | 15.46 | 94.56 |

Table 8: SimpleTOD results on MultiWOZ 2.1 using oracle information.

# C   Dialogue State Tracking Analysis

This section provides more detailed analysis of Dialogue State Tracking (DST) task performance by SimpleTOD, as mentioned in section 5.

**Understanding Long, Multi-domain Context**   Table 9 indicates the DST performance of SimpleTOD in case of multi-domain and long context. This example also shows understanding dialogue states, where slots across domains are related. For example, *taxi departure* and *taxi destination* should be inferred from *attraction name* and *hotel name*.

| | |
|---|---|
| **Context ('MUL1015', turn 10)** | \<|context|\> \<|user|\> i am looking for a pool somewhere in the south of cambridge . \<|system|\> i am sorry , but it does not look like we have a pool that matches your criteria . \<|user|\> how about some type of theater ? \<|system|\> the junction theatre is in the south area . would you like the address ? \<|user|\> yes , please provide the address and postcode as well . \<|system|\> the junction theatre is located on clifton way at postcode cb17gx . is there anything else i can assist you with ? \<|user|\> thanks . yes , i also need to find a guesthouse for my stay there . \<|system|\> are you needing the guesthouse in the south , also ? do you have a preference in price range ? \<|user|\> the area should be the centre . i need it to be cheap and have free parking . \<|system|\> i have 2 guesthouses that fit your criteria . 1 is the alexander bed and breakfast and the other is el shaddai . \<|user|\> what is the star rating -s for them ? \<|system|\> the alexander bed and breakfast has a 4 star rating while the el shaddai has 0 stars . would you like to book a room with either of these ? \<|user|\> ok great , thank you . i also need to book a taxi to get back and forth . \<|system|\> where did you wan to leave from and go to ? \<|user|\> i'll be leaving the theatre at 14:45 and going to the guesthouse . \<|system|\> let s back up a moment . which lodgings did you prefer , so we know where the taxi will be taking you ? \<|user|\> let s go with the alexander bed and breakfast . \<|system|\> your booking is complete , a grey volkswagen will be picking you up , contact number 07753429052 . can i help with anything else ? \<|user|\> cool beans ! sorry i was so much trouble . thanks so much for all of your help , you need a raise dealing with customers like me ! \<|endofcontext|\> |
| **Target** | \<|belief|\> attraction area south, attraction name the junction, attraction type theatre, hotel area centre, hotel name alexander bed and breakfast, hotel parking yes, hotel pricerange cheap, hotel type guesthouse, taxi departure the junction, taxi destination alexander bed and breakfast, taxi leaveat 14:45 \<|belief|\> \<|action|\> general bye none \<|endofaction|\> \<|response|\> it was no trouble at all ! i hope you have a fantastic trip and enjoy your stay ! \<|endofresponse|\> |
| **SimpleTOD** | \<|belief|\> attraction area south, attraction name the junction, attraction type theatre, hotel area centre, hotel name alexander bed and breakfast, hotel parking yes, hotel pricerange cheap, hotel type guesthouse, taxi departure the junction, taxi destination alexander bed and breakfast, taxi leaveat 14:45 \<|endofbelief|\> \<|action|\> general bye none \<|endofaction|\> \<|response|\> thank you for using our service \<|endofresponse|\> |

Table 9: Example of multi-domain dialogue state tracking for long context, where dialogue context spanning over 3 domains, i.e. attraction, hotel, and taxi. The belief states for taxi departure and destination (highlighted) should be inferred from hotel and attraction domains.

**Regarding Noisy Annotations**   Concrete examples of four types of noisy-labeled annotation defined in section 5, are shown in Tables 10, 11, 12, 13 and 14. The results indicate that SimpleTOD is robust to noisy annotation and can often generate the correct belief state in situations where annotation is incorrect. As mentioned in section 4.1, the list of noisy annotations is released with this paper.

**Full Dialogue Example**   Table 15 and Table 16 include several turns as part of a full dialogue for delexicalized and lexicalized outputs. These are from our strongest SimpleTOD model in the setting that requires it to generate all outputs, which means that it ignores all DB Search results.

| | |
|---|---|
| **Context ('SNG01608', turn 2)** | <\|context\|> <\|user\|> are there any portuguese restaurant -s in cambridge ? <\|system\|> yes there is a portuguese restaurant in cambridge with 2 different location -s , would you like the addresses ? <\|user\|> if 1 of them has a moderate price range please give me that address . if not tell me about turkish restaurant -s instead . <\|endofcontext\|> |
| **Target** | <\|belief\|> restaurant food turkish, restaurant pricerange moderate <\|endofbelief\|> |
| **SimpleTOD** | <\|belief\|> restaurant food portugese, restaurant pricerange moderate<\|endofbelief\|> |
| **Context ('MUL1575', turn 3)** | <\|context\|> <\|user\|> i would like to find a particular restaurant in cambridge . the name of the restaurant is restaurant 2 two . could you give me the location ? <\|system\|> restaurant 2 two is nice french restaurant located at 22 chesterton road chesterton . would like me to book you a table ? <\|user\|> that would be great . i need it for 8 on friday . <\|system\|> do you have a time preference ? <\|user\|> yes at 11:15 if that s not available i can do 10:15 <\|endofcontext\|> |
| **Target** | <\|belief\|> restaurant book day friday, restaurant book people 8, restaurant book time 10:15, restaurant name restaurant 2 two<\|endofbelief\|> |
| **SimpleTOD** | <\|belief\|> restaurant book day friday, restaurant book people 8, restaurant book time 11:15, restaurant name restaurant 2 two <\|endofbelief\|> |
| **Context ('PMUL1194', turn 6)** | <\|context\|> <\|user\|> hello , i need information about old schools . <\|system\|> there are 5 local colleges , do you have a particular area you are interested in ? <\|user\|> i do not have an area preference . <\|system\|> old schools is an architectural attraction in the center of town . they are located on trinity lane , in postcode cb21tt . <\|user\|> what is the phone number ? <\|system\|> the phone number is 01223332320 . do you need any more information ? <\|user\|> yes , i am looking for a train that leaves on wednesday for cambridge <\|system\|> i have over 200 entries . let s try to narrow that down . what time would you like to arrive ? <\|user\|> monday at 11:00 for 5 people please <\|system\|> just to confirm , you want to leave on monday , or wednesday ? and where are you departing from ? <\|user\|> leaving ely anytime after 15:15 and i need to know how long the ride is and the specific arrival time please . <\|endofcontext\|> |
| **Target** | <\|belief\|> attraction name old schools, train book people 5, train day monday, train departure ely, train destination cambridge, train leaveat 15:15 <\|endofbelief\|> 0 |
| **SimpleTOD** | <\|belief\|> attraction name old schools, train book people 5, train day wednesday, train departure ely, train destination cambridge, train leaveat 15:15 <\|endofbelief\|> |
| **Context ('MUL1024', turn 7)** | <\|context\|> <\|user\|> i am looking for a theatre to visit in the centre of cambridge . can you help me with this ? <\|system\|> i can recommend the adc theatre on park street or the cambridge arts theatre located at 6 st edward s passage <\|user\|> are either in the city center ? <\|system\|> both are located in the city centre . <\|user\|> can you give me the address of the adc theatre please ? <\|system\|> the address is park street . <\|user\|> what films are playing today ? <\|system\|> i am sorry . i do not have that information . can i help you with anything else ? <\|user\|> i am looking for a particular hotel called avalon . <\|system\|> avalon is located at 62 gilbert road . would you like me to book a reservation ? <\|user\|> great , can i book that for 6 people for 3 nights starting on tuesday . <\|system\|> i am sorry , the booking was unsuccessful . would you like to try for another date or a shorter stay ? <\|user\|> would you be interested in staying at the avalon for 1 night ? if not we can find another hotel available for a 3 night stay for 6 guests . <\|endofcontext\|> |
| **Target** | <\|belief\|> attraction area centre, attraction name adc theatre, attraction type theatre, hotel book day tuesday, hotel book people 6, hotel book stay 3, hotel name avalon <\|endofbelief\|> |
| **SimpleTOD** | <\|belief\|> attraction area centre, attraction name adc theatre, attraction type theatre, hotel book day tuesday, hotel book people 6, hotel book stay 1, hotel name avalon <\|endofbelief\|> |

Table 10: Examples of Type 1 noisy-labeled annotation, context lacks enough information to infer the true belief state.

| | |
|---|---|
| **Context ('MUL0088', turn 2)** | <\|context\|> <\|user\|> i am looking for a cheap hotel with free parking near cambridge . <\|system\|> i have multiple cheap hotel -s with free parking . what part of town are you interested in staying in ? <\|user\|> i would like to stay close to the center area , but the hotel should be 3 star . <\|endofcontext\|> |
| **Target** | <\|belief\|> hotel parking yes, hotel pricerange cheap <\|endofbelief\|> |
| **SimpleTOD** | <\|belief\|> hotel area centre, hotel parking yes, hotel pricerange cheap, hotel stars 3 <\|endofbelief\|> |
| **Context ('PMUL2437', turn 3)** | <\|context\|> <\|user\|> i want to find a moderate -ly priced restaurant . <\|system\|> i have many options available for you ! is there a certain area or cuisine that interests you ? <\|user\|> yes i would like the restaurant to be located in the center of the attractions . <\|system\|> there are 21 restaurant -s available in the centre of town . how about a specific type of cuisine ? <\|user\|> i need to know the food type and postcode and it should also have mutliple sports <\|endofcontext\|> |
| **Target** | <\|belief\|> restaurant area centre, restaurant pricerange moderate <\|endofbelief\|> |
| **SimpleTOD** | <\|belief\|> attraction type multiple sports, restaurant area centre, restaurant pricerange moderate <\|endofbelief\|> |
| **Context ('PMUL2437', turn 3)** | <\|context\|> <\|user\|> i want to find a moderate -ly priced restaurant . <\|system\|> i have many options available for you ! is there a certain area or cuisine that interests you ? <\|user\|> yes i would like the restaurant to be located in the center of the attractions . <\|system\|> there are 21 restaurant -s available in the centre of town . how about a specific type of cuisine ? <\|user\|> i need to know the food type and postcode and it should also have mutliple sports <\|endofcontext\|> |
| **Target** | <\|belief\|> restaurant area centre, restaurant pricerange moderate <\|endofbelief\|> |
| **SimpleTOD** | <\|belief\|> attraction type multiple sports, restaurant area centre, restaurant pricerange moderate <\|endofbelief\|> |
| **Context ('MUL1060', turn 4)** | <\|context\|> <\|user\|> hello , i would like to find a hotel that include -s free parking . <\|system\|> most of the hotel -s in town offer free parking . is there a certain area you would like to stay in , or do you have a price range in mind ? <\|user\|> yes . the centre would be nice and also free wifi . <\|system\|> the university arms is an expensive , 4 star hotel with free wifi . comparatively , the alexander bed and breakfast is a cheap -ly priced guesthouse , also 4 stars . <\|user\|> please book me some rooms for the university arms to accommodate 8 people for 3 nights starting on wednesday . can you also provide me the reference number after you book ? <\|system\|> your reference number is x5ny66zv . <\|user\|> thank you . can you please help me find a place to go in town in the same area as the hotel ? preferably a college . <\|endofcontext\|> |
| **Target** | <\|belief\|> attraction area centre, attraction name college, hotel area centre, hotel book day wednesday, hotel book people 8, hotel book stay 3, hotel name university arms hotel, hotel parking yes <\|endofbelief\|> |
| **SimpleTOD** | <\|belief\|> attraction area centre, attraction type college, hotel area centre, hotel book day wednesday, hotel book people 8, hotel book stay 3, hotel internet yes, hotel name university arms hotel, hotel parking yes <\|endofbelief\|> |
| **Context ('MUL1642', turn 5)** | <\|context\|> <\|user\|> hello , i am trying to find a train that goes from cambridge to london kings cross . can you help me book a ticket ? <\|system\|> i can help with that . can you tell me what day you will be traveling ? <\|user\|> i need to leave on saturday after 18:45 . <\|system\|> the soonest departure time would be at 19:00 on saturday , is that okay ? <\|user\|> yes , that s perfect . can you book that for 8 people ? <\|system\|> you are all booked with reference number 144vdbrm . the cost of 151.04 gbp will be payable at the station . can i be of further assistance today ? <\|user\|> i am looking for an expensive place to eat in the centre , what is there that fits that criteria ? <\|system\|> there 33 place -s that fit your criteria . do you have a particular cuisine type in mind so that i can narrow the results down ? <\|user\|> it does not matter what kind of food . what would you recommend for a large group of 8 people ? <\|endofcontext\|> |
| **Target** | <\|belief\|> restaurant area centre, restaurant pricerange expensive, train book people 8, train day saturday, train departure cambridge, train destination london kings cross, train leaveat 18:45 <\|endofbelief\|> |
| **SimpleTOD** | <\|belief\|> restaurant area centre, restaurant book people 8, restaurant pricerange expensive, train book people 8, train day saturday, train departure cambridge, train destination london kings cross, train leaveat 18:45 <\|endofbelief\|> |
| **Context ('MUL0088', turn 7)** | <\|context\|> <\|user\|> i am looking for a cheap hotel with free parking near cambridge . <\|system\|> i have multiple cheap hotel -s with free parking . what part of town are you interested in staying in ? <\|user\|> i would like to stay close to the center area , but the hotel should be 3 star . <\|system\|> we do not have any hotel -s that match your search . do you want to try something else ? <\|user\|> are there any moderate 3 star hotel -s with free parking ? <\|system\|> we do not have any hotel -s that match your search . do you want to try something else ? <\|user\|> how about 1 in the moderate price range ? <\|system\|> we have 3 entries that match your preferences . would you prefer north , south , or west ? <\|user\|> i do not have a preference but i would also like ot find a restaurant called the cow pizza kitchen and bar . <\|system\|> before i book your restaurant would you like to book your lodging ? i think you will like hamilton lodge . it meets your needs . <\|user\|> does it have internet ? <\|system\|> yes , the hamilton lodge has internet . <\|user\|> sounds great . what is the address and contact information ? <\|endofcontext\|> |
| **Target** | <\|belief\|> hotel internet yes, hotel parking yes, hotel pricerange moderate, hotel stars 3, restaurant name cow pizza kitchen and bar <\|endofbelief\|> |
| **SimpleTOD** | <\|belief\|> hotel internet yes, hotel name hamilton lodge, hotel parking yes, hotel pricerange moderate, hotel stars 3, restaurant name cow pizza kitchen and bar <\|endofbelief\|> |

Table 11: Examples of Type 2 noisy-labeled annotation, belief state is not labeled, while context contains the information.

| | |
|---|---|
| **Context ('MUL1926', turn 1)** | <\|context\|> <\|user\|> yes hello ! i am looking for ==a train leaving on tuesday== from norwich to cambridge . <\|endofcontext\|> |
| **Target** | <\|belief\|> train departure norwich, train destination cambridge <\|endofbelief\|> |
| **SimpleTOD** | <\|belief\|> ==train day tuesday,== train departure norwich, train destination cambridge <\|endofbelief\|> |
| **Context ('MUL1650', turn 3)** | <\|context\|> <\|user\|> i am looking for a chinese restaurant in the centre of cambridge . <\|system\|> there are quite a few chinese restaurant -s in the area ! do you have a price range you would like to stay within ? <\|user\|> no , anything is fine . can you suggest a great 1 ? <\|system\|> jinling noodle bar looks like a good place . <\|user\|> can i have their phone number and address ? ==also , are there any trains that go from peterborough to cambridge ?== <\|endofcontext\|> |
| **Target** | <\|belief\|> restaurant area centre, restaurant food chinese, restaurant name jinling noodle bar <\|endofbelief\|> |
| **SimpleTOD** | <\|belief\|> restaurant area centre, restaurant food chinese, restaurant name jinling noodle bar, ==train departure peterborough, train destination cambridge== <\|endofbelief\|> |
| **Context ('MUL2376', turn 3)** | <\|context\|> <\|user\|> where would i find the museum of archaeology and anthropology ? <\|system\|> the museum of archaelogy and anthropology is located at university of cambridge , downing street . their phone number is 01223333516 . do you need anything else ? <\|user\|> and the attraction type is a museum , right ? <\|system\|> yes , ==it is a museum== . <\|user\|> sounds great . thank you for the info ! <\|endofcontext\|> |
| **Target** | <\|belief\|> attraction name museum of archaelogy and anthropology <\|endofbelief\|> |
| **SimpleTOD** | <\|belief\|> attraction name museum of archaelogy and anthropology, ==attraction type museum== <\|endofbelief\|> |
| **Context ('MUL1076', turn 9)** | <\|context\|> <\|user\|> yes i am looking for some hotel -s that include free parking and cheap rates in the cambridge area . <\|system\|> there are 10 hotel -s , do you have any other preferences to narrow it down ? <\|user\|> sure . i would like a guesthouse . <\|system\|> there are 9 options available . do you have a preference for how many stars or what area specifically ? <\|user\|> i do not care . please book 1 that you recommend . <\|system\|> ok , ==how about the allenbell ?== to book a stay i'll need to know the first day you want to stay , how many days , and how many guests . <\|user\|> i need it for 8 people , and for 4 nights starting saturday . <\|system\|> great , i have booked your stay for 8 people for 4 nights . your reference number is pb41wuot . may i help with anything else ? <\|user\|> i also would like to find a museum in the west . <\|system\|> there are several museums there , any preferences ? <\|user\|> i would like to find the museum of science , please . <\|system\|> i am sorry , but i am having trouble locating your requested attraction . can you please restate your request ? <\|user\|> is there a museum of science ? <\|system\|> i am not finding 1 , sorry . <\|user\|> are there any other museums in that area ? <\|system\|> yes there are 7 museums in the west area of town for your pleasure . is there anything else you would like to narrow them down with ? <\|user\|> the museum i want should be in the west side of town and can i have the postcode for the museum . <\|endofcontext\|> |
| **Target** | <\|belief\|> attraction area west, attraction type museum, hotel book day saturday, hotel book people 8, hotel book stay 4, hotel parking yes, hotel pricerange cheap, hotel type guesthouse <\|endofbelief\|> |
| **SimpleTOD** | <\|belief\|> attraction area west, attraction type museum, hotel book day saturday, hotel book people 8, hotel book stay 4, ==hotel name allenbell,== hotel parking yes, hotel pricerange cheap, 'hotel type guesthouse <\|endofbelief\|> |

Table 12: Examples of Type 2 noisy-labeled annotation, belief state is not labeled, while context contains the information.

| | |
|---|---|
| **Context ('SNG1036', turn 1)** | \<\|context\|\> \<\|user\|\> i was not planning to stay in town tonight , but it looks like i am going to have to . can you help me find a pretty cheap room ? \<\|endofcontext\|\> |
| **Target** | \<\|belief\|\> hotel pricerange cheap, hotel type hotel \<\|endofbelief\|\> |
| **SimpleTOD** | \<\|belief\|\> hotel pricerange cheap \<\|endofbelief\|\> |
| **Context ('PMUL1194', turn 1)** | \<\|context\|\> \<\|user\|\> hello , i need information about old schools . \<\|endofcontext\|\> |
| **Target** | \<\|belief\|\> attraction type college \<\|endofbelief\|\> |
| **SimpleTOD** | \<\|belief\|\> attraction name old schools \<\|endofbelief\|\> |
| **Context ('SNG0284', turn 2)** | \<\|context\|\> \<\|user\|\> i want a train from london liverpool street arriving at 9:30 \<\|system\|\> which day will you be traveling ? \<\|user\|\> i want to leave on thursday . |
| **Target** | \<\|belief\|\> train arriveby 09:30, train book people 2, train day thursday, train departure london liverpool street \<\|endofbelief\|\> |
| **SimpleTOD** | \<\|belief\|\> train arriveby 09:30, train day thursday, train departure london liverpool street |
| **Context ('PMUL0069', turn 2)** | \<\|context\|\> \<\|user\|\> i am looking for some hungarian food restaurant -s near the centre , please . \<\|system\|\> i am sorry there are no hungarian restaurant -s near centre . \<\|user\|\> what kind of expensive restaurant -s are in the center of town ? |
| **Target** | \<\|belief\|\> restaurant area centre, restaurant food hungarian, restaurant pricerange expensive \<\|endofbelief\|\> |
| **SimpleTOD** | \<\|belief\|\> restaurant area centre, restaurant pricerange expensive \<\|endofbelief\|\> |
| **Context ('PMUL3688', turn 3)** | \<\|context\|\> \<\|user\|\> i am looking for a place to go in the centre of town . \<\|system\|\> is there any type of attraction you would like to see ? \<\|user\|\> any of your choice . get me the address and entrance fee \<\|system\|\> the holy trinity church is located on market street . the entrance fee is free . is there anything else i can help with ? \<\|user\|\> i also need a train out to the airport on friday . \<\|endofcontext\|\> |
| **Target** | \<\|belief\|\> attraction area centre, train day friday, train destination stansted airport\<\|endofbelief\|\> |
| **SimpleTOD** | \<\|belief\|\> attraction area centre, train day friday \<\|endofbelief\|\> |

Table 13: Examples of Type 3 noisy-labeled annotation, belief state is labeled, while context lacks the information.

| | |
|---|---|
| **Context ('SNG02207', turn 1)** | \<\|context\|\> \<\|user\|\> i need to book a tax departing from gandhi . \<\|endofcontext\|\> |
| **Target** | \<\|belief\|\> taxi departure the gandhi \<\|endofbelief\|\> |
| **SimpleTOD** | \<\|belief\|\> taxi departure gandhi \<\|endofbelief\|\> |
| **Context ('MUL1811', turn 2)** | \<\|context\|\> \<\|user\|\> i am looking for something to do in town . are there any interesting attraction in the north part of town ? \<\|system\|\> yes , there are several places: i see a boat , a park , and 2 swimming pools . would 1 of these work for you ? \<\|user\|\> ooh , a pool would be fun . can you give me their contact info ? \<\|endofcontext\|\> |
| **Target** | \<\|belief\|\> attraction area north, attraction type pool \<\|endofbelief\|\> |
| **SimpleTOD** | \<\|belief\|\> attraction area north, attraction type swimming pool \<\|endofbelief\|\> |
| **Context ('PMUL0548', turn 4)** | \<\|context\|\> \<\|user\|\> hello , i would like some information on a restaurant . \<\|system\|\> do you have a certain cuisine or area in mind ? \<\|user\|\> i want british food . in the east , please . \<\|system\|\> the graffton hotel restaurant is in the east and serves british food . it s location is 619 newmarket rd . fen ditton . \<\|user\|\> i will need it for saturday , 7 people at 17:45 please . \<\|system\|\> i have successfully booked a table for 7 on saturday at 17:45 for you . the table will be reserved for 15 minutes and your reference number is r8cv0hu9 . \<\|user\|\> i am also looking for a train on sunday . \<\|endofcontext\|\> |
| **Target** | \<\|belief\|\> restaurant area east, restaurant book day saturday, restaurant book people 7, restaurant book time 17:45, restaurant food british, restaurant name grafton hotel restaurant, train day sunday \<\|endofbelief\|\> |
| **SimpleTOD** | \<\|belief\|\> restaurant area east, restaurant book day saturday, restaurant book people 7, restaurant book time 17:45, restaurant food british, restaurant name graffton hotel restaurant, train day sunday \<\|endofbelief\|\> |
| **Context ('PMUL0550', turn 7)** | \<\|context\|\> \<\|user\|\> hey i am looking for a train from cambridge to bishops stortford . mind helping out ? \<\|system\|\> many trains are leaving on friday . mind choosing the time ? \<\|user\|\> i want to leave on monday and arrive by 18:45 . \<\|system\|\> i have train tr4283 that leaves cambridge at 5:29 and arrives in bishops stortford at 6:07 . would you like to make reservations ? \<\|user\|\> could you give me the travel time and price of that train please ? \<\|system\|\> the cost is 10.10 pounds . the duration of the trip is 38 minutes , leaving cambridge at 5:29 and arriving at 6:07 in bishops stortford . \<\|user\|\> i also want a cheap chinese restaurant . \<\|system\|\> there are 4 great restaurant -s to choose from . do you prefer 1 on the south side of town or in the centre ? \<\|user\|\> i would like to be in the centre please . \<\|system\|\> i recommend charlie chan . would you like to reserve a table ? \<\|user\|\> yes . monday , 8 people , 10:30 . \<\|system\|\> booking was successful . the table will be reserved for 15 minutes . reference number is : ifjmbkli . \<\|user\|\> thank you very much . that will be all for today . i appreciate all your help . \<\|endofcontext\|\> |
| **Target** | \<\|belief\|\> restaurant area centre, restaurant book day monday, restaurant book people 8, restaurant book time 10:30, restaurant food chinese, restaurant name charlie, restaurant pricerange cheap, train arriveby 18:45, train day monday, train departure cambridge, train destination bishops stortford \<\|endofbelief\|\> |
| **SimpleTOD** | \<\|belief\|\> restaurant area centre, restaurant book day monday, restaurant book people 8, restaurant book time 10:30, restaurant food chinese, restaurant name charlie chan, restaurant pricerange cheap, train arriveby 18:45, train day monday, train departure cambridge, train destination bishops stortford \<\|endofbelief\|\> |

Table 14: Examples of Type 4 noisy-labeled annotation, belief state value is misspelled.

| | | |
|---|---|---|
| **Turn 1** | Context | <\|context\|> <\|user\|> hello , i need a train to ely on sunday please <\|endofcontext\|> |
| | Target | <\|belief\|> train destination ely <\|endofbelief\|> <\|action\|> train inform choice, train inform day, train inform dest, train request arrive <\|endofaction\|> <\|response\|> we have [value_count] trains traveling to [value_place] on [value_day] . what time would you like to arrive ? <\|endofresponse\|> |
| | SimpleTOD | <\|belief\|> train day sunday, train destination ely <\|endofbelief\|> <\|action\|> train inform choice, train inform day, train inform leave, train inform depart, train inform dest, train inform arrive, train inform id, train offerbook none <\|endofaction\|> <\|response\|> there are [value_count] trains leaving [value_place] on [value_day].　[train_id] leaves [value_place] at [value_time] and arrives in [value_place] at [value_time]. would you like to book a ticket? <\|endofresponse\|> |
| **Turn 2** | Context | <\|context\|> <\|user\|> hello , i need a train to ely on sunday please <\|system\|> we have 10 trains traveling to ely on sunday . what time would you like to arrive ?  <\|user\|> how about 1 arriving by 13:00 departing from cambridge . <\|endofcontext\|> |
| | Target | <\|belief\|> train destination ely, train day sunday, train arriveby 13:00, train departure cambridge <\|endofbelief\|> <\|action\|> train inform arrive, train inform day, train inform dest, train inform id, train offerbook none <\|endofaction\|> <\|response\|> [train_id] will get you to [value_place] by [value_time] on [value_day] . should i book you a ticket ? <\|endofresponse\|> |
| | SimpleTOD | <\|belief\|> train day sunday, train destination ely, train departure cambridge, train arriveby 13:00 <\|endofbelief\|> <\|action\|> train inform arrive, train inform leave, train inform id, train offerbook none <\|endofaction\|> <\|response\|> [train_id] leaves at [value_time] and arrives at [value_time]. would like me to book it for you? <\|endofresponse\|> |
| **Turn 3** | Context | <\|context\|> <\|user\|> hello , i need a train to ely on sunday please <\|system\|> we have 10 trains traveling to ely on sunday . what time would you like to arrive ?  <\|user\|> how about 1 arriving by 13:00 departing from cambridge . <\|system\|> tr1159 will get you to ely by 12:07 on sunday . should i book you a ticket ? <\|user\|> yes , that would be great . i'll need tickets for 8 people . do you have a reference number ? <\|endofcontext\|> |
| | Target | <\|belief\|> train destination ely, train day sunday, train arriveby 13:00, train departure cambridge, train book people 8 <\|endofbelief\|> <\|action\|> train offerbooked ref, train offerbooked ticket <\|endofaction\|> <\|response\|> booking was successful , the total fee is [value_price] gbp payable at the station . reference number is [train_reference] . <\|endofresponse\|> |
| | SimpleTOD | <\|belief\|> train book people 8, train arriveby 13:00, train day sunday, train departure cambridge, train destination ely <\|endofbelief\|> <\|action\|> general reqmore none, train offerbooked ref, train offerbooked ticket <\|endofaction\|> <\|response\|> booking was successful, the total fee is [value_price] gbp payable at the station. reference number is [train_reference]. is there anything else i can help you with? <\|endofresponse\|> |
| **Turn 4** | Context | <\|context\|> <\|user\|> hello , i need a train to ely on sunday please <\|system\|> we have 10 trains traveling to ely on sunday . what time would you like to arrive ? <\|user\|> how about 1 arriving by 13:00 departing from cambridge . <\|system\|> tr1159 will get you to ely by 12:07 on sunday . should i book you a ticket ? <\|user\|> yes , that would be great . i'll need tickets for 8 people . do you have a reference number ? <\|system\|> booking was successful , the total fee is 28.16 gbp payable at the station . reference number is : hdx8ix63 . <\|user\|> i am also looking for a museum . can you recommend 1 ? <\|endofcontext\|> |
| | Target | <\|belief\|> train book people 8, train arriveby 13:00, attraction type museum, train day sunday, train departure cambridge, train destination ely <\|endofbelief\|> <\|action\|> attraction recommend name, general greet none <\|endofaction\|> <\|response\|> how about [attraction_name] ? it looks lovely <\|endofresponse\|> |
| | SimpleTOD | <\|belief\|> train book people 8, train arriveby 13:00, attraction type museum, train day sunday, train departure cambridge, train destination ely <\|endofbelief\|> <\|action\|> attraction recommend name, attraction recommend area, attraction recommend fee, attraction recommend addr, attraction recommend phone <\|endofaction\|> <\|response\|> i would recommend the [attraction_name] in the [value_area]. it is located at [attraction_address]. the entrance fee is [value_count] pounds. the phone number is [attraction_phone] <\|endofresponse\|> |
| **Turn 5** | Context | <\|context\|> <\|user\|> hello , i need a train to ely on sunday please <\|system\|> we have 10 trains traveling to ely on sunday . what time would you like to arrive ? <\|user\|> how about 1 arriving by 13:00 departing from cambridge . <\|system\|> tr1159 will get you to ely by 12:07 on sunday . should i book you a ticket ? <\|user\|> yes , that would be great . i'll need tickets for 8 people . do you have a reference number ? <\|system\|> booking was successful , the total fee is 28.16 gbp payable at the station . reference number is : hdx8ix63 . <\|user\|> i am also looking for a museum . can you recommend 1 ? <\|system\|> how about cafe jello gallery ? it looks lovely <\|user\|> that will work . could you get me the phone number and area its in ? <\|endofcontext\|> |
| | Target | <\|belief\|> train book people 8, train arriveby 13:00, attraction type museum, train day sunday, train departure cambridge, train destination ely <\|endofbelief\|> <\|action\|> general greet none, general reqmore none, attraction inform phone, attraction inform area <\|endofaction\|> <\|response\|> sure !  it s in the [value_area] area of town , and the phone number is [attraction_phone] . can i help you with anything else ? <\|endofresponse\|> |
| | SimpleTOD | <\|belief\|> train book people 8, train arriveby 13:00, attraction type museum, train day sunday, train departure cambridge, train destination ely <\|endofbelief\|> <\|action\|> attraction inform phone, attraction inform area, attraction inform name <\|endofaction\|> <\|response\|> [attraction_name] is in the [value_area] area and the phone number is [attraction_phone] <\|endofresponse\|> |
| **Turn 6** | Context | <\|context\|> <\|user\|> hello , i need a train to ely on sunday please <\|system\|> we have 10 trains traveling to ely on sunday . what time would you like to arrive ? <\|user\|> how about 1 arriving by 13:00 departing from cambridge . <\|system\|> tr1159 will get you to ely by 12:07 on sunday . should i book you a ticket ? <\|user\|> yes , that would be great . i'll need tickets for 8 people . do you have a reference number ? <\|system\|> booking was successful , the total fee is 28.16 gbp payable at the station . reference number is : hdx8ix63 . <\|user\|> i am also looking for a museum . can you recommend 1 ? <\|system\|> how about cafe jello gallery ? it looks lovely <\|user\|> that will work . could you get me the phone number and area its in ? <\|system\|> sure ! it s in the west area of town , and the phone number is 01223312112 . can i help you with anything else ? <\|user\|> you were great today . thanks so much for all the help . that is all . goodbye . <\|endofcontext\|> |
| | Target | <\|belief\|> train book people 8, train arriveby 13:00, attraction type museum, train day sunday, train departure cambridge, train destination ely <\|endofbelief\|> <\|action\|> general bye none, general welcome none, general greet none <\|endofaction\|> <\|response\|> it was a pleasure to help .  have a good day . goodbye . <\|endofresponse\|> |
| | SimpleTOD | <\|belief\|> train book people 8, train arriveby 13:00, attraction type museum, train day sunday, train departure cambridge, train destination ely <\|endofbelief\|> <\|action\|> general bye none <\|endofaction\|> <\|response\|> thank you for using our service <\|endofresponse\|> |

Table 15: SimpleTOD end-to-end generation (delexicalized) on MultiWOZ (dialogue 'PMUL3663').

| | | |
|---|---|---|
| **Turn 1** | Context | \<\|context\|\> \<\|user\|\> hello , i need a train to ely on sunday please \<\|endofcontext\|\> |
| | Target | \<\|belief\|\> train destination ely \<\|endofbelief\|\> \<\|action\|\> train inform choice, train inform day, train inform dest, train request arrive \<\|endofaction\|\> \<\|response\|\> we have 10 trains traveling to ely on sunday . what time would you like to arrive ? \<\|endofresponse\|\> |
| | SimpleTOD | \<\|belief\|\> train day sunday, train destination ely \<\|endofbelief\|\> \<\|action\|\> train inform choice, train inform day, train inform leave, train inform depart, train inform dest, train inform arrive, train inform id, train offerbook none \<\|endofaction\|\> \<\|response\|\> there are 10 trains leaving cambridge on sunday. tr3177 leaves cambridge at 9:50 and arrives in ely at 10:07. would you like to book a ticket? \<\|endofresponse\|\> |
| **Turn 2** | Context | \<\|context\|\> \<\|user\|\> hello , i need a train to ely on sunday please \<\|system\|\> we have 10 trains traveling to ely on sunday . what time would you like to arrive ? \<\|user\|\> how about 1 arriving by 13:00 departing from cambridge . \<\|endofcontext\|\> |
| | Target | \<\|belief\|\> train destination ely, train day sunday, train arriveby 13:00, train departure cambridge \<\|endofbelief\|\> \<\|action\|\> train inform arrive, train inform day, train inform dest, train inform id, train offerbook none \<\|endofaction\|\> \<\|response\|\>tr1159 will get you to ely by 12:07 on sunday . should i book you a ticket ? \<\|endofresponse\|\> |
| | SimpleTOD | \<\|belief\|\> train day sunday, train destination ely, train departure cambridge, train arriveby 13:00 \<\|endofbelief\|\> \<\|action\|\> train inform arrive, train inform leave, train inform id, train offerbook none \<\|endofaction\|\> \<\|response\|\> tr1159 leaves at 11:50 and arrives at 12:07. would you like me to book it for you? \<\|endofresponse\|\> |
| **Turn 3** | Context | \<\|context\|\> \<\|user\|\> hello , i need a train to ely on sunday please \<\|system\|\> we have 10 trains traveling to ely on sunday . what time would you like to arrive ? \<\|user\|\> how about 1 arriving by 13:00 departing from cambridge . \<\|system\|\> tr1159 will get you to ely by 12:07 on sunday . should i book you a ticket ? \<\|user\|\> yes , that would be great . i'll need tickets for 8 people . do you have a reference number ? \<\|endofcontext\|\> |
| | Target | \<\|belief\|\> train destination ely, train day sunday, train arriveby 13:00, train departure cambridge, train book people 8 \<\|endofbelief\|\> \<\|action\|\> train offerbooked ref, train offerbooked ticket \<\|endofaction\|\> \<\|response\|\>booking was successful , the total fee is 28.16 gbp payable at the station . reference number is hdx8ix63. \<\|endofresponse\|\> |
| | SimpleTOD | \<\|belief\|\> train book people 8, train arriveby 13:00, train day sunday, train departure cambridge, train destination ely \<\|endofbelief\|\> \<\|action\|\> general reqmore none, train offerbooked ref, train offerbooked ticket \<\|endofaction\|\> \<\|response\|\> booking was successful, the total fee is 28.16 gbp payable at the station. reference number is hdx8ix63. is there anything else i can help you with? \<\|endofresponse\|\> |
| **Turn 4** | Context | \<\|context\|\> \<\|user\|\> hello , i need a train to ely on sunday please \<\|system\|\> we have 10 trains traveling to ely on sunday . what time would you like to arrive ? \<\|user\|\> how about 1 arriving by 13:00 departing from cambridge . \<\|system\|\> tr1159 will get you to ely by 12:07 on sunday . should i book you a ticket ? \<\|user\|\> yes , that would be great . i'll need tickets for 8 people . do you have a reference number ? \<\|system\|\> booking was successful , the total fee is 28.16 gbp payable at the station . reference number is : hdx8ix63 . \<\|user\|\> i am also looking for a museum . can you recommend 1 ? \<\|endofcontext\|\> |
| | Target | \<\|belief\|\> train book people 8, train arriveby 13:00, attraction type museum, train day sunday, train departure cambridge, train destination ely \<\|endofbelief\|\> \<\|action\|\> attraction recommend name, general greet none \<\|endofaction\|\> \<\|response\|\> how about cafe jello gallery ? it looks lovely \<\|endofresponse\|\> |
| | SimpleTOD | \<\|belief\|\> train book people 8, train arriveby 13:00, attraction type museum, train day sunday, train departure cambridge, train destination ely \<\|endofbelief\|\> \<\|action\|\> attraction recommend name, attraction recommend area, attraction recommend fee, attraction recommend addr, attraction recommend phone \<\|endofaction\|\> \<\|response\|\> i would recommend the cafe jello gallery in the west. it is located at 13 magdalene street. the entrance fee is 0 pounds. the phone number is 01223464646 \<\|endofresponse\|\> |
| **Turn 5** | Context | \<\|context\|\> \<\|user\|\> hello , i need a train to ely on sunday please \<\|system\|\> we have 10 trains traveling to ely on sunday . what time would you like to arrive ? \<\|user\|\> how about 1 arriving by 13:00 departing from cambridge . \<\|system\|\> tr1159 will get you to ely by 12:07 on sunday . should i book you a ticket ? \<\|user\|\> yes , that would be great . i'll need tickets for 8 people . do you have a reference number ? \<\|system\|\> booking was successful , the total fee is 28.16 gbp payable at the station . reference number is : hdx8ix63 . \<\|user\|\> i am also looking for a museum . can you recommend 1 ? \<\|system\|\> how about cafe jello gallery ? it looks lovely \<\|user\|\> that will work . could you get me the phone number and area its in ? \<\|endofcontext\|\> |
| | Target | \<\|belief\|\> train book people 8, train arriveby 13:00, attraction type museum, train day sunday, train departure cambridge, train destination ely \<\|endofbelief\|\> \<\|action\|\> general greet none, general reqmore none, attraction inform phone, attraction inform area \<\|endofaction\|\> \<\|response\|\> sure ! it s in the west area of town , and the phone number is 01223464646 . can i help you with anything else ? \<\|endofresponse\|\> |
| | SimpleTOD | \<\|belief\|\> train book people 8, train arriveby 13:00, attraction type museum, train day sunday, train departure cambridge, train destination ely \<\|endofbelief\|\> \<\|action\|\> attraction inform phone, attraction inform area, attraction inform name \<\|endofaction\|\> \<\|response\|\> cafe jello gallery is in the west area and the phone number is 01223464646 \<\|endofresponse\|\> |
| **Turn 6** | Context | \<\|context\|\> \<\|user\|\> hello , i need a train to ely on sunday please \<\|system\|\> we have 10 trains traveling to ely on sunday . what time would you like to arrive ? \<\|user\|\> how about 1 arriving by 13:00 departing from cambridge . \<\|system\|\> tr1159 will get you to ely by 12:07 on sunday . should i book you a ticket ? \<\|user\|\> yes , that would be great . i'll need tickets for 8 people . do you have a reference number ? \<\|system\|\> booking was successful , the total fee is 28.16 gbp payable at the station . reference number is : hdx8ix63 . \<\|user\|\> i am also looking for a museum . can you recommend 1 ? \<\|system\|\> how about cafe jello gallery ? it looks lovely \<\|user\|\> that will work . could you get me the phone number and area its in ? \<\|system\|\> sure ! it s in the west area of town , and the phone number is 01223312112 . can i help you with anything else ? \<\|user\|\> you were great today . thanks so much for all the help . that is all . goodbye . \<\|endofcontext\|\> |
| | Target | \<\|belief\|\> train book people 8, train arriveby 13:00, attraction type museum, train day sunday, train departure cambridge, train destination ely \<\|endofbelief\|\> \<\|action\|\> general bye none, general welcome none, general greet none \<\|endofaction\|\> \<\|response\|\> it was a pleasure to help . have a good day . goodbye . \<\|endofresponse\|\> |
| | SimpleTOD | \<\|belief\|\> train book people 8, train arriveby 13:00, attraction type museum, train day sunday, train departure cambridge, train destination ely \<\|endofbelief\|\> \<\|action\|\> general bye none \<\|endofaction\|\> \<\|response\|\> thank you for using our service \<\|endofresponse\|\> |

Table 16: SimpleTOD end-to-end generation (lexicalized) on MultiWOZ (dialogue 'PMUL3663').