[Reviews · NeurIPS 2020]

Review 1

Summary and Contributions: This paper is about using a unified language model to build task-oriented dialog systems

Strengths: The paper has compared with many recent dialog work.

Weaknesses: Mostly is the innovation is not that huge. It is hard to see how much improvement this is compared to Sololist and ARDM

Correctness: The method looks correct

Clarity: Yes

Relation to Prior Work: Missing reference: SOLOIST: Few-shot Task-Oriented Dialog with A Single Pre-trained Auto-regressive Model Qingyang Wu, Yichi Zhang, Yu Li, and Zhou Yu. 2019. Alternating recurrent dialog model with large-scale pre-trained language models

Reproducibility: Yes

Additional Feedback:


Review 2

Summary and Contributions: This paper proposed a single end-to-end model SimpleTOD for task-oriented dialog by leveraging existing pre-trained models. Empirical results on the MultiWOZ dataset demonstrated that the proposed methods had outperformed all prior methods for DST tasks and dialog act/response generation in the e2e setting.

Strengths: Leveraging pre-trained models to solve problems in task-oriented dialogues has become a trend in recent years. By taking advantage of pre-trained models, this paper achieves SOTA performance in the DST task on the MultiWOZ dataset.

Weaknesses: • This paper did not mention SOLOIST, which has achieved SOTA in policy optimization and the end2end setting for MultiWOZ 2.0, as illustrated in the MultiWOZ repository.  • Although SimpleTOD produces SOTA results for the DST task on MultiWOZ, the gain over the previous SOTA model is small.  • No human evaluation is provided for response generation. • Since there are existing prior and concurrent works that leverage pre-trained models like GPT-2, based on my understanding, the primary differentiator (novelty) in this paper is limited. • If there is an experiment on datasets other than MultiWOZ, the paper's quality will be better.

Correctness: The claims and methods seem correct.

Clarity: This paper is well written in general. But I think more discussions should be made to compare the approach with other papers that employ pre-trained models.

Relation to Prior Work: [6][17] uses similar idea as this paper, but the differences are not well discussed. Besides, the paper 'SOLOIST' was never mentioned in the paper.

Reproducibility: Yes

Additional Feedback: ------------------------------------------------------------------------------------------ A response after the rebuttal period: After reading authors' reply, I agree that aforementioned papers are considered concurrent work. With human evaluation added in the final version as the authors mentioned, I think this work should be good.


Review 3

Summary and Contributions: Post AR: I see the other reviewers mostly are concerned with novelty in the face of SOLOIST and DAMD. I emphasize the contemporaneousness of SOLOIST and feel satisfied with discussion of DAMD. I still very much want to see this published. The authors consider the task of task-oriented dialogue on multiwoz, where an assistant needs to help a user find a train/restaurant/whatever in the city center of Cambridge. The authors propose a fully end-to-end model, in which the entire task is structured as sequence prediction using a transformer. The authors report state of the art results on multiwoz, while previous end-to-end models tend to perform worse. The authors provide some analysis of several aspects of it (special tokens, pretraining ablation, oracle belief state, etc) to help clarify source of the results. Their model is also conceptually much simpler than most of the prior attempts.

Strengths: I found the work quite compelling, and would like to see it published. With respect to Ham et al., the analysis and state of the art results are primary areas of novelty, but these two works MAY be considered contemporary, and so they may be viewed as sharing the core contribution of applying GPT-2 to task oriented dialogue for both response generation AND belief state tracking. One particularly strong aspect of this paper is that I read it on arxiv a month or two ago, and our lab was interested in the results, so we internally replicated them. We found that we got similar numbers without too much effort, and that the results carried over with other pretraining sources too. To this end, the paper is more sound than the VAST majority of NeurIPS papers: in a short amount of time, a third party was able to replicate the results using only the description from the paper.

Weaknesses: - Only use one dataset, but many task oriented datasets exist. Another dataset would strengthen this model a bit. - Weak differences with Ham et al. or Soloist. All three works should probably be perceived as contemporaneous, and I don't think should detract from the novelty of this work, as they came out within 30 days of each other. I would rather the authors include detailed explanations to help explain the primary differences, for the benefit of future readers.

Correctness: Yes. As emphasized in Strengths, my lab has even replicated these results, indicating that they are very robust.

Clarity: Yes, I find the paper mostly easy to read. The explanation of MultiWoZ may be unclear to someone unfamiliar with the details of the dataset, especially around some of the aspects of the database results, etc. The authors may consider finding a way to make Figure 1 more central in the text to make this part clear. The authors would also benefit greatly from putting Figure 2 of the appendix in the main text, as that makes the special tokens analysis much easier to understand.

Relation to Prior Work: The only problematic one is the differentiation with Ham et al., which does pretty much the same thing, but got worse results. The authors should more clearly delineate the differences between their work, and attempt to explain the performance differences. Otherwise, prior work is reasonably covered. They drop some potential related work, (for example, https://arxiv.org/abs/1809.01984 pretrained on reddit for dialogue a year before Henderson did), but the bibliography is reasonable.

Reproducibility: Yes

Additional Feedback: I need to say that I read this paper on arxiv a month or two before reviewing it for NeurIPS, so this is not a double blind review. However, I feel it's a good paper, and that my lab has internally reproduced it already lends it strong credibility. I feel that the authors would benefit from dropping section 3.3 and using the extra space to include a copy of the Figure 2 from the appendix.


Review 4

Summary and Contributions: The authors propose SimpleTOD, which can replace modular task-oriented dialogue models to unified causal language model in an end-2-end manner. There are three sub-tasks in the task-oriented dialogue. They are dialogue state tracking, action prediction, and response generation. SimpleTOD treats all three sub-tasks as sequence generation. Whole up to dialogue context C_t is used as the first input to the model, and the model generates dialogue state B_t at turn t. The dialogue state B_t is used for database search. The DB search results are (domain, slot, value) triplets, but they use aggregated result D_t that includes the number of rows satisfying the conditions and booking information. And then, the model receives concatenated C_t, B_t, and D_t as input to generate action A_t. At last, all the conditions are combined to generate system response S_t. The experimental results show that SimpleTOD achieves state-of-the-art performance in the MultiWOZ dataset. Especially their model is evaluated below various settings regarding noisy label-cleaning in the dialogue state tracking task. They argue their model is robust to noisy annotations. In action and response generation benchmark, It also achieves better performance than previous SOTA models. They emphasize the importance of usage of oracle information in the end-2-end evaluation. So, They compare their model with only one model DAMD, which doesn’t use oracle information except for DB search results. Additional analysis shows interesting observations. Especially the usage of special tokens to the input is a very important factor in this case.

Strengths: They propose a simple unified framework to solve task-oriented dialogue with an end-to-end manner. It is convincing all sub-tasks are cascaded continuously and exchange positive signals to each other. From this point of view, their choice on causal language modeling as their framework is very natural and a good point. It could reduce efforts to train each component for task-oriented dialogue. They evaluate their model on truly end-2-end setting without any oracle information for the first time. The model achieves state-of-the-art performance in the multi-domain task-oriented dialogue dataset.

Weaknesses: Their contribution lies in simplicity. It is good applications however fundamentally not novel. For more convincing results, The author should argue the modular pipeline problems and show the effectiveness of the end-to-end model compared with a seperated model more precisely. With this comparison, I want to know more about the advantages of multi-task learning approach and correlations between the sub-tasks since they insist on inherent dependencies between the sub-tasks. However, this analysis is missing decisively.

Correctness: Their method is simple and clear.

Clarity: The overall description of the causal language model is well written. However, its formal representation regarding DB search result D_t and each special tokens are ambiguous. It is confusing that they described their own transformer models too simple, but their main results are based on the pre-trained GPT-2 model (There are differences between these models e.g., activation function (ReLU vs. Gelu), position encoding (sinusoidal vs. learned)).

Relation to Prior Work: There is a lack of explanation for the baseline model. For fair comparison regarding robustness on the noisy annotation in the dialogue state tracking, they should show it on other models too. When it comes to end-to-end evaluation, they described it well except for precise analysis on the oracle information.

Reproducibility: Yes

Additional Feedback: Regarding oracle database search, I don’t fully understand their analysis. Especially DAMD+augmentation outperforms other models when it uses two oracle information compared with not using that information in Table 7. Even though there are noisy annotations to search result, It is not a scalable setting to expand new domains or new DB instances. I think it needs more explorations why SimpleTOD model is vulnerable from the oracle information compared with other baseline models in Table 7. It would have been better if the inference speed and performance trade-off were also compared when choosing pre-trained model. It is Because autoregressive generation has time-consuming natures, especially with long sequence length. ------------------------------------------------------------------------------------------ A response after the rebuttal period: Thank you for responding to the review. I have agreed that it is impressive to show that a simplified input scheme with a simple generative model outperforms all previous models on DST. I believe the paper is worthy of being reported to NeurIPS community. ------------------------------------------------------------------------------------------

[Author Response · NeurIPS 2020]

We thank the reviewers for the constructive feedback. Following are response to some of the comments.

**Innovation is not huge (R1, R2, R4)** The novelty of the proposed SimpleTOD is based on simplifying task-oriented dialogue as a causal language modeling. Moreover, we show that using a simplified input sequence definition, special tokens, and embedding layer, the model can achieve higher performance compared to all previous models on DST. It also achieved state-of-the-art results on end-to-end setting, without any additional pretraining (used in SOLOIST) or multi-action data augmentation (used by DAMD).

**Missing references (R1, R2)** ARDM is only cited in the experiment section, line 205, and compared in the Table 7 (oracle information) in appendix B. SOLOIST appeared on arxiv after our paper, and we assumed it's a concurrent work. Detailed description of ARDM and SOLOIST will be added in related section and Tables in the camera ready version.

**Weak difference compared to previous pretrained models (R2, R3)** The Soloist model and model proposed by Ham et al. [17] are appeared within 30 days of ours. **ARDM** is pretrained model in alternating role for user and system, without using belief state or action annotation. It achieved better results than its predecessor on context-to-response generation (dialogue policy), but it requires oracle belief states to compute inform and success rate and is not applicable to end-to-end evaluation. It is also not designed for DST task. **Ham et al. [17]** used pretrained model trained on serialized sequence of dialogue. As mentioned in our paper, the role of special tokens are crucial in achieving better performance. They employed delimiter tokens for different segments, i.e. <usr>, <sys>, <ds> and <sa>. However, our tokenization is different than delimiter tokens, and is based on choosing semantic words which help the model to learn the semantic of each segments, i.e. belief, action, and response, and end of segment tokens such as <|endofbelief|>. Additionally, they used token-type embedding for user and system sequences, where our model does not use this embedding. Their proposed model achieved lower performance than previous baselines for DST and end-to-end evaluation, whereas we outperforms all previous models. **Pawel et al. [7]** used a different approach for creating input sequence by prepending dialogue context with belief state and DB search results. Their evaluation is only for context-to-response generation, and is not applicable to end-to-end setting. **SOLOIST** model removes action from input sequence and also used token-type embedding. It is pretrained on seven more dialogue datasets, before finetuning on MultiWOZ. Moreover, they used a data augmentation during training (contrastive learning), similar to DAMD paper, where they combined dialogue context and belief with a negative response and used the final token for classification, to improve their end-to-end performance. However, they did not report end-to-end performance without pretraining on other dialogue datasets. Their "w/o pretraining" setting is only evaluated on low resource training. Also, they did not report DST performance. **In summary, our proposed model is much simpler, in terms of (1) input sequence definition, (2) embedding layers, (3) training algorithm, and (4) pretraining, which make it easy to reproduce the results, and outperformed previous models on DST and end-to-end setting.**

**Human evaluation (R2)**: An experiment with human conversing with the model in multi-domain setting to complete the task is conducted. This evaluation will be added in the camera ready version with a live demo.

**Small Performance gain on DST (R2)**: Previous models on DST task have used bidirectional encoder and mostly based on BERT pretrained model. DSTQA, SST, and TripPy also used additional supervisions. However, our proposed model indicates that a unidirectional encoder can achieve state of the art results with no extra supervision. All previous models use a label cleaning, where our model outperforms them without cleaning, according to Table 1. The analysis shows that our model is robust to noisy annotation, especially for Type 2, known as early markup.

**Experiment on other datasets (R2, R3, R4):** MultoWOZ is the largest dataset that all previous models are evaluated for both DST and end-to-end settings. DAMD is not evaluated on CamRest767, which is a small dataset, and no previous model is evaluated on Schema-guided dialogue (SGD). Therefore, we focus our evaluation on MultiWOZ for a fair comparison. Evaluation on other dataset such as SGD and CamRest676 will be added in the camera-ready version.

**Comparison with modular-based models (R4)** We run an experiment similar to modular-based setting three language models for belief, action, and response generation, which briefly mentioned in line 218-222. Based on experiment, separate models can improve individual scores, but the combined score remains nearly identical.

**Vulnerability to oracle information (R4)** As shown in Table 7 of Appendix B, DAMD achieved higher performance when using oracle information. We suspect that due to augmentation, where they exploit single-to-many mapping of dialogue, that combined context and belief states with different true action and responses, they gained a higher performance. This is due to better action/response generation. However, DAMD performs lower in end–to-end setting perhaps due to lower accuracy in generating belief states that affect the subsequent tasks (action/response generation)

**Analysis on inference speed (R4):** This section will be added to the camera-ready version, with comparison to other DST and end-to-end methods. Moreover, the previous models on DST should classify or generate value for all slots, which is not scalable to large domain-slot situation. However, autoregressive models only generate slot-values that exist in dialogue context.

[Meta-Review · NeurIPS 2020]

All reviewers find this work quite strong both in terms of approach and results, and reviewers applaud that the work has proved robustly reproducible. One important point is that several contemporaneous papers share some commonalities with this submission. We agree that they were published less than a month before the deadline and should therefore be considered contemporaneous; however it would have been much better scientific practice to include the discussion of these works in the submission, if the authors were aware of them -- regardless on when the authors put their initial submission on arxiv. The discussion that situates the submission in the context of these other works in the authors response is enlightening and interesting, and should definitely be in the final version. Conditioned on this being the case, we are happy to accept the paper. Authors should also use the additional page in the final version to incorporate the many excellent suggestions from reviewers to improve the paper.